HACSim: an R package to estimate intraspecific sample sizes for genetic diversity assessment using haplotype accumulation curves

Phillips Jarrett D. jphill01@uoguelph.ca 1
French Steven H. 1
Hanner Robert H. 2
Gillis Daniel J. 1
1 School of Computer Science, University of Guelph , Guelph , Ontario , Canada
2 Department of Integrative Biology, Biodiversity Institute of Ontario, University of Guelph , Guelph , Ontario , Canada
Mei Gang
Electronic publication date: 2020 Jan 6
Publication date: 2020
Volume: 6
Electronic Location ID: e243
Received 2019 Jul 18; Accepted 2019 Nov 7
Copyright: ©2020 Phillips et al.
Copyright year: 2020
Copyright holder: Phillips et al.
License: This is an open access article distributed under the terms of the Creative Commons Attribution License, which permits unrestricted use, distribution, reproduction and adaptation in any medium and for any purpose provided that it is properly attributed. For attribution, the original author(s), title, publication source (PeerJ Computer Science) and either DOI or URL of the article must be cited.
License URL: https://creativecommons.org/licenses/by/4.0/

Keywords: Algorithm, DNA barcoding, Extrapolation, Iterative method, Sampling sufficiency, Species

Funding: College of Physical and Engineering Science (CPES) Graduate Excellence Entrance Scholarship This work was supported by the College of Physical and Engineering Science (CPES) Graduate Excellence Entrance Scholarship to Jarrett D. Phillips. The funders had no role in study design, data collection and analysis, decision to publish, or preparation of the manuscript.

==============================
Assessing levels of standing genetic variation within species requires a robust sampling for the purpose of accurate specimen identification using molecular techniques such as DNA barcoding; however, statistical estimators for what constitutes a robust sample are currently lacking. Moreover, such estimates are needed because most species are currently represented by only one or a few sequences in existing databases, which can safely be assumed to be undersampled. Unfortunately, sample sizes of 5–10 specimens per species typically seen in DNA barcoding studies are often insufficient to adequately capture within-species genetic diversity. Here, we introduce a novel iterative extrapolation simulation algorithm of haplotype accumulation curves, called HACSim (Haplotype Accumulation Curve Simulator) that can be employed to calculate likely sample sizes needed to observe the full range of DNA barcode haplotype variation that exists for a species. Using uniform haplotype and non-uniform haplotype frequency distributions, the notion of sampling sufficiency (the sample size at which sampling accuracy is maximized and above which no new sampling information is likely to be gained) can be gleaned. HACSim can be employed in two primary ways to estimate specimen sample sizes: (1) to simulate haplotype sampling in hypothetical species, and (2) to simulate haplotype sampling in real species mined from public reference sequence databases like the Barcode of Life Data Systems (BOLD) or GenBank for any genomic marker of interest. While our algorithm is globally convergent, runtime is heavily dependent on initial sample sizes and skewness of the corresponding haplotype frequency distribution.

Introduction

Background

Earth is in the midst of its sixth mass extinction event and global biodiversity is declining at an unprecedented rate (Ceballos et al., 2015). It is therefore important that species genetic diversity be catalogued and preserved. One solution to address this mounting crisis in a systematic, yet rapid way is DNA barcoding (Hebert et al., 2003). DNA barcoding relies on variability within a small gene fragment from standardized regions of the genome to identify species, based on the fact that most species exhibit a unique array of barcode haplotypes that are more similar to each other than those of other species (e.g., a barcode “gap”). In animals, the DNA barcode region corresponds to a 648 bp fragment of the 5′ terminus of the cytochrome c oxidase subunit I (COI) mitochondrial marker (Hebert et al., 2003; Hebert, Ratnasingham & De Waard, 2003). A critical problem since the inception of DNA barcoding involves determining appropriate sample sizes necessary to capture the majority of existing intraspecific haplotype variation for major animal taxa (Hebert et al., 2004; Meyer & Paulay, 2005; Ward et al., 2005). Taxon sample sizes currently employed in practice for rapid assignment of a species name to a specimen, have ranged anywhere from 1–15 specimens per species (Matz & Nielsen, 2005; Ross, Murugan & Li, 2008; Goodall-Copestake, Tarling & Murphy, 2012; Jin, He & Zhang, 2012; Yao et al., 2017); however, oftentimes only 1–2 individuals are actually collected. This trend is clearly reflected within the Barcode of Life Data Systems (http://www.boldsystems.org) (Ratnasingham & Hebert, 2007), where an overwhelming number of taxa have only a single record and sequence.

A fitting comparison to the issue of adequacy of specimen sample sizes can be made to the challenge of determining suitable taxon distance thresholds for species separation on the basis of the DNA barcode gap (Meyer & Paulay, 2005). It has been widely demonstrated that certain taxonomic groups, such as Lepidoptera (butterflies/moths), are able to be readily separated into distinct clusters largely reflective of species boundaries derived using morphology (Čandek & Kuntner, 2015). However, adoption of a fixed limit of 2% difference between maximum intraspecific distance and minimum interspecific (i.e., nearest-neighbour) divergence is infeasible across all taxa (Hebert, Ratnasingham & De Waard, 2003; Collins & Cruickshank, 2013). Species divergence thresholds should be calculated from available sequence data obtained through deep sampling of taxa across their entire geographic ranges whenever possible (Young et al., 2017). There is a clear relationship between specimen sample sizes and observed barcoding gaps: sampling too few individuals can give the impression of taxon separation, when in fact none exists  (Meyer & Paulay, 2005; Hickerson, Meyer & Moritz, 2006; Wiemers & Fiedler, 2007; Dasmahapatra et al., 2010; Čandek & Kuntner, 2015), inevitably leading to erroneous conclusions (Collins & Cruickshank, 2013). It is thus imperative that barcode gap analyses be based on adequate sample sizes to minimize the presence of false positives. Introducing greater statistical rigour into DNA barcoding appears to be the clear way forward in this respect (Nielsen & Matz, 2006; Čandek & Kuntner, 2015; Luo et al., 2015; Phillips, Gillis & Hanner, 2019). The introduction of computational approaches for automated species delimitation such as Generalized Mixed Yule Coalescent (GMYC) (Pons et al., 2006; Monaghan et al., 2009; Fujisawa & Barraclough, 2013), Automatic Barcode Gap Discovery (ABGD) (Puillandre, Lambert & Brouillet, 2011) and Poisson Tree Processes (PTP) (Zhang et al., 2013) has greatly contributed to this endeavour in the form of web servers (GMYC, ABGD, PTP) and R packages (GMYC: Species’ LImits by Threshold Statistics, splits (Ezard, Fujisawa & Barraclough, 2017)).

Various statistical resampling and population genetic methods, in particular coalescent simulations, for the estimation of sample sizes, have been applied to Lepidoptera (Costa Rican skipper butterflies (Astraptes fulgerator)) (Zhang et al., 2010) and European diving beetles (Agabus bipustulatus) (Bergsten et al., 2012). Using Wright’s equilibrium island model (Wright, 1951) and Kimura’s stepping stone model (Kimura & Weiss, 1964) under varying effective population sizes and migration rates, Zhang et al. (2010) found that between 156-1985 specimens per species were necessary to observe 95% of all estimated COI variation for simulated specimens of A. fulgerator. Conversely, real species data showed that a sample size of 250-1188 individuals is probably needed to capture the majority of COI haplotype variation existing for this species (Zhang et al., 2010). A subsequent investigation carried out by Bergsten et al. (2012) found that a random sample of 250 individuals was required to uncover 95% COI diversity in A. bipustulatus; whereas, a much smaller sample size of 70 specimens was necessary when geographic separation between two randomly selected individuals was maximized.

Others have employed more general statistical approaches. Based on extensive simulation experiments, through employing the Central Limit Theorem (CLT), Luo et al. (2015) suggested that no fewer than 20 individuals per species be sampled. Conversely, using an estimator of sample size based on the Method of Moments, an approach to parameter estimation relying on the Weak Law of Large Numbers (Pearson, 1894), sample sizes ranging from 150–5,400 individuals across 18 species of ray-finned fishes (Chordata: Actinopterygii) were found by Phillips et al. (2015).

Haplotype accumulation curves paint a picture of observed standing genetic variation that exists at the species level as a function of expended sampling effort (Phillips et al., 2015; Phillips, Gillis & Hanner, 2019). Haplotype sampling completeness can then be gauged through measuring the slope of the curve, which gives an indication of the number of new haplotypes likely to be uncovered with additional specimens collected. For instance, a haplotype accumulation curve for a hypothetical species having a slope of 0.01 suggests that only one previously unseen haplotype will be captured for every 100 individuals found. This is strong evidence that the haplotype diversity for this species has been adequately sampled. Thus, further recovery of specimens of such species provide limited returns on the time and money invested to sequence them. Trends observed from generated haplotype accumulation curves for the 18 actinopterygian species assessed by Phillips et al. (2015), which were far from reaching an asymptote, corroborated the finding that the majority of intraspecific haplotypes remain largely unsampled in Actinopterygii for even the best-represented species in BOLD. Estimates obtained from each of these studies stand in sharp contrast to sample sizes typically reported within DNA barcoding studies.

Numerical optimization methods are required to obtain reasonable approximations to otherwise complex questions. Many such problems proceed via the iterative method, whereby an initial guess is used to produce a sequence of successively more precise (and hopefully more accurate) approximations. Such an approach is attractive, as resulting solutions can be made as precise as desired through specifying a given tolerance cutoff. However, in such cases, a closed-form expression for the function being optimized is known a priori. In many instances, the general path (behaviour) of the search space being explored is the only information known, and not its underlying functional form. In this paper, we take a middle-ground approach that is an alternative to probing sampling completeness on the basis of haplotype accumulation curve slope measurement. To this end, iteration is applied to address the issue of relative sample size determination for DNA barcode haplotype sampling completeness, a technique suggested by Phillips, Gillis & Hanner (2019). Given that specimen collection and processing is quite a laborious and costly endeavour (Cameron, Rubinoff & Will, 2006; Stein et al., 2014), the next most direct solution to an otherwise blind search strategy is to employ computational simulation that approximates specimen collection in the field. The main contribution of this work is the introduction of a new, easy-to-use R package implementing a novel statistical optimization algorithm to estimate sample sizes for assessment of genetic diversity within species based on saturation observed in haplotype accumulation curves. Here, we present a novel nonparametric stochastic (Monte Carlo) iterative extrapolation algorithm for the generation of haplotype accumulation curves based on the approach of Phillips et al. (2015). Using the statistical environment R (R Core Team, 2018), we examine the effect of altering species haplotype frequencies on the shape of resulting curves to inform on likely required sample sizes needed for adequate capture of within-species haplotype variation. Proof-of-concept of our method is illustrated through both hypothetical examples and real DNA sequence data.

Motivation

Consider N DNA sequences that are randomly sampled for a given species of interest across its known geographic range, each of which correspond to a single specimen. Suppose further that H* of such sampled DNA sequences are unique (i.e., are distinct haplotypes). This scenario leads naturally to the following question: What is N*, the estimated total number of DNA sequence haplotypes that exist for a species θ? Put another way, what sample size (number of specimens) is needed to capture the existing haplotype variation for a species?

The naïve approach (adopted by Phillips et al. (2015)) would be to ignore relative frequencies of observed haplotypes; that is, assume that species haplotypes are equally probable in a species population. Thus, in the absence of any information, the best one can do is adopt a uniform distribution for the number of sampled haplotypes. Such a path leads to obtaining gross overestimates for sufficient sampling (Phillips et al., 2015). A much better approach uses all available haplotype data to arrive at plausible estimates of required taxon sample sizes. This latter method is explored here in detail.

Methods

Haplotype accumulation curve simulation algorithm

Algorithm functions

Our algorithm, HACSim (short for Haplotype Accumulation Curve Simulator), consisting of two user-defined R functions, HAC.sim() and HAC.simrep(), was created to run simulations of haplotype accumulation curves based on user-supplied parameters. The simulation treats species haplotypes as distinct character labels relative to the number of individuals possessing a given haplotype. The usual convention in this regard is that Haplotype 1 is the most frequent, Haplotype 2 is the next most frequent, etc. (Gwiazdowski et al., 2013). A haplotype network represents this scheme succinctly (Fig. 1).

Figure 1 Modified haplotype network from Phillips, Gillis & Hanner (2019).

Haplotypes are labelled according to their absolute frequencies such that the most frequent haplotype is labelled “1”, the second-most frequent haplotype is labelled “2”, etc., and is meant to illustrate that much species locus variation consists of rare haplotypes at very low frequency (typically only represented by 1 or 2 specimens). Thus, species showing such patterns in their haplotype distributions are probably grossly under-respresented in public sequence databases like BOLD and GenBank.

Such an implementation closely mimics that seen in natural species populations, as each character label functions as a unique haplotype linked to a unique DNA barcode sequence. The algorithm then randomly samples species haplotype labels in an iterative fashion with replacement until all unique haplotypes have been observed. This process continues until all species haplotypes have been sampled. The idea is that levels of species haplotypic variation that are currently catalogued in BOLD can serve as proxies for total haplotype diversity that may exist for a given species. This is a reasonable assumption given that, while estimators of expected diversity are known (e.g., Chao1 abundance) (Chao, 1984), the frequencies of unseen haplotypes are not known a priori. Further, assuming a species is sampled across its entire geographic range, haplotypes not yet encountered are presumed to occur at low frequencies (otherwise they would likely have already been sampled).

Because R is an interpreted programming language (i.e., code is run line-by-line), it is slow compared to faster alternatives which use compilation to convert programs into machine-readable format; as such, to optimize performance of the present algorithm in terms of runtime, computationally-intensive parts of the simulation code were written in the C++ programming language and integrated with R via the packages Rcpp (Eddelbuettel & François, 2011) and RcppArmadillo (Eddelbuettel & Sanderson, 2014). This includes function code to carry out haplotype accumulation (via the function accumulate(), which is not directly called by the user). A further reason for turning to C++ is because some R code (e.g., nested ‘for’ loops) is not easily vectorized, nor can parallelization be employed for speed improvement due to loop dependence. The rationale for employing R for the present work is clear: R is free, open-source software that it is gaining widespread use within the DNA barcoding community due to its ease-of-use and well-established user-contributed package repository (Comprehensive R Archive Network (CRAN)). As such, the creation and disemination of HACSim as a R framework to assess levels of standing genetic variation within species is greatly facilitated.

A similar approach to the novel one proposed here to automatically generate haplotype accumulation curves from DNA sequence data is implemented in the R package spider (SPecies IDentity and Evolution in R; (Brown et al., 2012)) using the haploAccum() function. However, the approach, which formed the basis of earlier work carried out by Phillips et al. (2015), is quite restrictive in its functionality and, to our knowledge, is currently the only method available to generate haplotype accumulation curves in R because spider generates haplotype accumulation curves from DNA sequence alignments only and is not amenable to inclusion of numeric inputs for specimen and haplotype numbers. Thus, the method could not be easily extended to address our question. This was the primary reason for the proposal of a statistical model of sampling sufficiency by Phillips et al. (2015) and its extension described herein.

Algorithm parameters

At present, the algorithm (consisting of HAC.sim() and HAC.simrep()) takes 13 arguments as input (Table 1).

Table 1 Parameters inputted (first 7) and outputted (last six) by HAC.sim() and HAC.simrep(), along with their definitions.

Range refers to plausible values that each parameter can assume within the haplotype accumulation curve simulation algorithm. [ and ] indicate that a given value is included in the range interval; whereas, ( and ) indicate that a given value is excluded from the range interval. Simulation progress can be tracked through setting progress = TRUE within HACHypothetical() or HACReal(). Users can optionally specify that a file be created containing all information outputted to the R console (via the argument filename, which can be named as the user wishes).

Parameter	Definition	Range	
N	total number of specimens/DNA sequences	(1, ∞)	
H*	total number of unique haplotypes	(1, N]	
probs	haplotype probability distribution vector	(0, 1)	
p	proportion of haplotypes to recover	(0, 1]	
perms	total number of permutations	(1, ∞)	
input.seqs	analyze FASTA file of species DNA sequences	TRUE, FALSE	
conf.level	desired confidence level for confidence interval calculation	(0, 1)	
H	cumulative mean number of haplotypes sampled	[1, H∗]	
H∗ − H	cumulative mean number of haplotypes not sampled	[0, H∗)	
R=HH∗	cumulative mean fraction of haplotypes sampled	(0, 1]	
H∗−HH∗	cumulative mean fraction of haplotypes not sampled	[0, 1)	
N∗	mean specimen sample size corresponding to H∗	[ N, ∞)	
N∗ − N	mean number of individuals not sampled	[0, N]	

A user must first specify the number of observed specimens/DNA sequences (N) and the number of observed haplotypes (i.e., unique DNA sequences) (H*) for a given species. Both N and H* must be greater than one. Clearly, N must be greater than or equal to H*.

Next, the haplotype frequency distribution vector must be specified. The probs argument allows for the inclusion of both common and rare species haplotypes according to user interest (e.g., equally frequent haplotypes, or a single dominant haplotype). The resulting probs vector must have a length equal to H*. For example, if H* = 4, probs must contain four elements. The total probability of all unique haplotypes must sum to one.

The user can optionally input the fraction of observed haplotypes to capture p. By default, p = 0.95, mirroring the approach taken by both Zhang et al. (2010) and Bergsten et al. (2012) who computed intraspecific sample sizes needed to recover 95% of all haplotype variation for a species. At this level, the generated haplotype accumulation curve reaches a slope close to zero and further sampling effort is unlikely to uncover any new haplotypes. However, a user may wish to obtain sample sizes corresponding to different haplotype recovery levels, e.g.,  p = 0.99 (99% of all estimated haplotypes found). In the latter scenario, it can be argued that 100% of species haplotype variation is never actually achieved, since with greater sampling effort, additional haplotypes are almost surely to be found; thus, a true asymptote is never reached. In any case, simulation completion times will vary depending on inputted parameter values, such as probs, which controls the skewness of the observed haplotype frequency distribution.

The perms argument is in place to ensure that haplotype accumulation curves “smooth out” and tend to H* asymptotically as the number of permutations (replications) is increased. The effect of increasing the number of permutations is an increase in statistical accuracy and consequently, a reduction in variance. The proposed simulation algorithm outputs a mean haplotype accumulation curve that is the average of perms generated haplotype accumulation curves, where the order of individuals that are sampled is randomized. Each of these perms curves is a randomized step function (a sort of random walk), generated according to the number of haplotypes found. A permutation size of 1,000 was used by Phillips et al. (2015) because smaller permutation sizes yielded non-smooth (noisy) curves. Permutation sizes larger than 1,000 typically resulted in greater computation time, with no noticeable change in accumulation curve behaviour (Phillips et al., 2015). By default, perms = 10,000 (in contrast to Phillips et al. (2015)), which is comparable to the large number of replicates typically employed in statistical bootstrapping procedures needed to ensure accuracy of computed estimates (Efron, 1979). Sometimes it will be necessary for users to sacrifice accuracy for speed in the presence of time constraints. This can be accomplished through decreasing perms. Doing so however will result in only near-optimal solutions for specimen sample sizes. In some cases, it may be necessary to increase perms to further smooth out the curves (to ensure monotonicity), but this will increase algorithm runtime substantially.

Should a user wish to analyze their own intraspecific COI DNA barcode sequence data (or sequence data from any single locus for that matter), setting input.seqs = TRUE allows this (via the read.dna() function in ape). In such a case, a pop-up file window will prompt the user to select the formatted FASTA file of aligned/trimmed sequences to be read into R. When this occurs, arguments for N, H* and probs are set automatically by the algorithm via functions available in the R packages ape (Analysis of Phylogenetics and Evolution) (Paradis, Claude & Strimmer, 2004) and pegas (Population and Evolutionary Genetics Analysis System) (Paradis, 2010). Users must be aware however that the number of observed haplotypes treated by pegas (via the haplotype() function) may be overestimated if missing/ambiguous nucleotide data are present within the final alignment input file. Missing data are explicitly handled by the base.freq() function in the ape package. When this occurs, R will output a warning that such data are present within the alignment. Users should therefore consider removing sequences or sites comprising missing/ambiguous nucleotides. This step can be accomplished using external software such as MEGA (Molecular Evolutionary Genetics Analysis; (Kumar, Stecher & Tamura, 2016)). The BARCODE standard (Hanner, 2009) was developed to help identify high quality sequences and can be used as a quality filter if desired. Exclusion of low-quality sequences also has the advantage of speeding up compution time of the algorithm significantly.

Options for confidence interval (CI) estimation and graphical display of haplotype accumulation is also available via the argument conf.level, which allows the user to specify the desired level of statistical confidence. CIs are computed from the sample α2100% and 1−α2100% quantiles of the haplotype accumulation curve distribution. The default is conf.level = 0.95, corresponding to a confidence level of 95%. High levels of statistical confidence (e.g., 99%) will result in wider confidence intervals; whereas low confidence leads to narrower interval estimates.

How does HACSim work?

Haplotype labels are first randomly placed on a two-dimensional spatial grid of size perms × N (read perms rows by N columns) according to their overall frequency of occurrence (Fig. 2).

Figure 2 Schematic of the HACSim optimization algorithm (setup, initialization and iteration).

Shown is a hypothetical example for a species mined from a biological sequence database like BOLD or GenBank with N = 5 sampled specimens (DNA sequences) possessing H* = 5 unique haplotypes. Each haplotype has an associated numeric ID from 1-H* (here, 1-5). Haplotype labels are randomly assigned to cells on a two-dimensional spatial array (ARRAY) with perms rows and N columns. All haplotypes occur with a frequency of 20%, (i.e., probs = (1/5, 1/5, 1/5, 1/5, 1/5)). Specimen and haplotype information is then fed into a black box to iteratively optimize the likely required sample size (N*) needed to capture a proportion of at least p haplotypes observed in the species sample.

The cumulative mean number of haplotypes is then computed along each column (i.e., for every specimen). If all H* haplotypes are not observed, then the grid is expanded to a size of perms × N* and the observed haplotypes enumerated. Estimation of specimen sample sizes proceeds iteratively, in which the current value of N* is used as a starting value to the next iteration (Fig. 2). An analogy here can be made to a game of golf: as one aims towards the hole and hits the ball, it gets closer and closer to the hole; however, one does not know the number of times to hit the ball before it lands in the hole. It is important to note that since sample sizes must be whole values, estimates of N* found at each iteration are rounded up to the next whole number. Even though this approach is quite conservative, it ensures that estimates are adequately reflective of the population from which they were drawn. HAC.sim(), which is called internally from HAC.simrep(), performs a single iteration of haplotype accumulation for a given species. In the case of real species, resulting output reflects current levels of sampling effort found within BOLD (or another similar sequence repository such as GenBank) for a given species. If the desired level of haplotype recovery is not reached, then HAC.simrep() is called to perform successive iterations until the observed fraction of haplotypes captured (R) is at least p. This stopping criterion is the termination condition necessary to halt the algorithm as soon as a “good enough” solution has been found. Such criteria are widely employed within numerical analysis. At each step of the algorithm, a dataset, in the form of a dataframe (called “d”) consisting of the mean number of haplotypes recovered (called means), along with the estimated standard deviation (sds) and the number of specimens sampled (specs) is generated. The estimated required sample size (N*) to recover a given proportion of observed species haplotypes corresponds to the endpoint of the accumulation curve. An indicator message is additionally outputted informing a user as to whether or not the desired level of haplotype recovery has been reached. The algorithm is depicted in Fig. 3.

Figure 3 Iterative extrapolation algorithm pseudocode for the computation of taxon sampling sufficiency employed within HACSim.

A user must input N, H* and probs to run simulations. Other function arguments required by the algorithm have default values and are not necessary to be inputted unless the user wishes to alter set parameters.

In Fig. 3, all input parameters are known a priori except Hi, which is the number of haplotypes found at each iteration of the algorithm, and Ri=HiH∗, which is the observed fraction of haplotype recovery at iteration i. The equation to compute N* (1) Ni+1∗=Ni+NiHiH∗−Hi=NiH∗Hi=NiRi

is quite intuitive since as Hi approaches H*, H∗ − Hi approaches zero, Ri=HiH∗ approaches one, and consequently, Ni approaches N*. In the first part of the above equation, the quantity NiHiH∗−Hi is the amount by which the haplotype accumulation curve is extrapolated, which incorporates random error and uncertainty regarding the true value of θ in the search space being explored. Nonparametric estimates formed from the above iterative method produce a convergent monotonically-increasing sequence, which becomes closer and closer to N* as the number of iterations increase; that is, (2) N1∗≤N2∗≤...≤Ni∗≤Ni+1∗→N∗

which is clearly a desirable property. Since haplotype accumulation curves are bounded below by one and bounded above by H*, then the above sequence has a lower bound equal to the initial guess for specimen sampling sufficiency (N) and an upper bound of N*.

Along with the iterated haplotype accumulation curves and haplotype frequency barplots, simulation output consists of the five initially proposed “measures of sampling closeness”, the estimate of θ (N*) based on Phillips et al. (2015)’s sampling model, in addition to the number of additional samples needed to recover all estimated total haplotype variation for a given species (N∗ − N; Fig. 4) (Table 1).

Figure 4 Graphical depiction of the iterative extrapolation sampling model as described in detail herein.

The figure is modified from Phillips, Gillis & Hanner (2019). The x-axis is meant to depict the number of specimens sampled, whereas the y-axis is meant to convey the cumulative number of unique haplotypes uncovered for every additional individual that is randomly sampled. Ni and Hi refer respectively to specimen and haplotype numbers that are observed at each iteration ( i) of HACSim for a given species. N* is the total sample size that is needed to capture all H* haplotypes that exist for a species.

These five quantities are given as follows: (1) Mean number of haplotypes sampled: Hi, (2) Mean number of haplotypes not sampled: H∗ − Hi, (3) Proportion of haplotypes sampled: HiH∗, (4) Proportion of haplotypes not sampled: H∗−HiH∗, (5) Mean number of individuals not sampled: N∗−Ni=NiHiH∗−Hi and are analogous to absolute and relative approximation error metrics seen in numerical analysis. It should be noted that the mean number of haplotypes captured at each iteration, Hi, will not necessary be increasing, even though estimates of the cumulative mean value of N* are. It is easily seen above that Hi approaches H* with increasing number of iterations. Similarly, as the simulation progresses, H∗ − Hi, H∗−HiH∗ and N∗−Ni=NiHiH∗−Hi all approach zero, while HiH∗ approaches one. The rate at which curves approach H* depends on inputs to both HAC.sim() and HAC.simrep(). Once the algorithm has converged to the desired level of haplotype recovery, a summary of findings is outputted consisting of (1) the initial guess (N) for sampling sufficiency; (2) the total number of iterations until convergence and simulation runtime (in seconds); (3) the final estimate (N*) of sampling sufficiency, along with an approximate (1 − α)100% confidence interval (see next paragraph); and, (4) the number of additional specimens required to be sampled (N∗ − N) from the initial starting value. Iterations are automatically separated by a progress meter for easy visualization.

An approximate symmetric (1 − α)100% CI for θ is derived using the (first order) Delta Method (Casella & Berger, 2002). This approach relies on the asymptotic normality result of the CLT and employs a first-order Taylor series expansion around θ to arrive at an approximation of the variance (and corresponding standard error) of N*. Such an approach is convenient since the sampling distribution of N* would likely be difficult to compute exactly due to specimen sample sizes being highly taxon-dependent. An approximate (large sample) (1 − α)100% CI for θ is given by (3) N∗±z1−α2σ ˆHHN∗

where z1−α2 denotes the appropriate critical value from the standard Normal distribution and σ ˆH is the estimated standard deviation of the mean number of haplotypes recovered at N*. The interval produced by this approach is quite tight, shrinking as Hi tends to H*. By default, HACSim computes 95% confidence intervals for the abovementioned quantities.

It is important to consider how a confidence interval for θ should be interpreted. For instance, a 95% CI for θ of (L, U), where L and U are the lower and upper endpoints of the confidence interval respectively, does not mean that the true sampling sufficiency lies between (L, U) with 95% probability. Instead, resulting confidence intervals for θ are themselves random and should be interpreted in the following way: with repeated sampling, one can be (1 − α)100% confident that the true sampling sufficency for p% haplotype recovery for a given species lies in the range (L, U) (1 − α)100% of the time. That is, on average, (1 − α)100% of constructed confidence intervals will contain θ (1 − α)100% of the time. It should be noted however that as given computed confidence intervals are only approximate in the limit, desired nominal probability coverage may not be achieved. In other words, the proportion of times calculated (1 − α)100% intervals actually contain θ may not be met.

HACSim has been implemented as an object-oriented framework to improve modularity and overall user-friendliness. Scenarios of hypothetical and real species are contained within helper functions which comprise all information necessary to run simulations successfully without having to specify certain function arguments beforehand. To carry out simulations of sampling haplotypes from hypothetical species, the function HACHypothetical() must first be called. Similarly, haplotype sampling for real species is handled by the function HACReal(). In addition to all input parameters rquired by HAC.sim() and HAC.simrep() outlined in Table 1, both HACHypothetical() and HACReal() take further arguments. Both functions take the optional argument filename which is used to save results outputted to the R console to a CSV file. When either HACHypothetical() or HACReal() is invoked (i.e., assigned to a variable), an object herein called HACSObj is created containing the 13 arguments employed by HACSim in running simulations. Note the generated object can have any name the user desires. Further, all simulation variables are contained in an environment called ‘envr’ that is hidden from the user.

Results

Here, we outline some simple examples that highlight the overall functionality of HACSim. When the code below is run, outputted results will likely differ from those depicted here since our method is inherently stochastic. Hence, it should be stressed that there is not one single solution for the problem at hand, but rather multiple solutions (Spall, 2012). This is in contrast to a completely deterministic model, where a given input always leads to the same unique output. To ensure reproducibility, the user can set a random seed value using the base R function set.seed() prior to running HAC.simrep(). It is important that a user set a working directory in R prior to running HACSim, which will ensure all created files (‘seqs.fas’ and ‘output.csv’) are stored in a single location for easy access and reference at a later time. In all scenarios, default parameters were unchanged (perms = 10,000, p = 0.95).

Application of HACSim to hypothetical species

Equal haplotype frequencies

Figure 5 shows sample graphical output of the proposed haplotype accumulation curve simulation algorithm for a hypothetical species with N = 100 and H* = 10. All haplotypes are assumed to occur with equal frequency (i.e., probs = 0.10). Algorithm output is shown below.

## Set parameters for hypothetical species ##	
> N<-100 # total number of sampled individuals	
> Hstar<-10 # total number of haplotypes	
> probs<-rep(1/Hstar, Hstar) # equal haplotype frequency	
### Run simulations ###	
> HACSObj<-HACHypothetical(N = N, Hstar = Hstar, probs = probs) # call helper function	
# set seed here if desired, e.g., set.seed(12345)	
> HAC.simrep(HACSObj)	
Simulating haplotype accumulation...	
—===============================================================================— 100%	
—Measures of Sampling Closeness —	
Mean number of haplotypes sampled: 10	
Mean number of haplotypes not sampled: 0	
Proportion of haplotypes sampled: 1	
Proportion of haplotypes not sampled: 0	
Mean value of N*: 100	
Mean number of specimens not sampled: 0	
Desired level of haplotype recovery has been reached	
———- Finished. ———-	
The initial guess for sampling sufficiency was N = 100 individuals	
The algorithm converged after 1 iterations and took 3.637 s	
The estimate of sampling sufficiency for p = 95% haplotype recovery is N* = 100 individuals ( 95% CI: 100-100 )	
The number of additional specimens required to be sampled for p = 95% haplotype recovery is N* - N = 0 individuals	

Algorithm output shows that R = 100% of the H* = 10 haplotypes are recovered from the random sampling of N = 100 individuals, with lower and upper 95% confidence limits of 100–100. No additional specimens need to be collected (N∗ − N = 0). Simulation results, consisting of the six “measures of sampling closeness” computed at each iteration, can be optionally saved in a comma-separated value (CSV) file called ‘output.csv’ (or another filename of the user’s choosing). Figure 5 shows that when haplotypes are equally frequent in species populations, corresponding haplotype accumulation curves reach an asymptote very quickly. As sampling effort is increased, the confidence interval becomes narrower, thereby reflecting one’s increased confidence in having likely sampled the majority of haplotype variation existing for a given species. Expected counts of the number of specimens possessing a given haplotype can be found from running max(envr$d$specs) * envr$probs in the R console once a simulation has converged. However, real data suggest that haplotype frequencies are not equal.

Figure 5 Graphical output of HAC.sim() for a hypothetical species with equal haplotype frequencies.

(A) Iterated haplotype accumulation curve. (B) Corresponding haplotype frequency barplot. For the generated haplotype accumulation curve, the 95% confidence interval for the number of unique haplotypes accumulated is depicted by gray error bars. Dashed lines depict the observed number of haplotypes (i.e., RH*) and corresponding number of individuals sampled found at each iteration of the algorithm. The dotted line depicts the expected number of haplotypes for a given haplotype recovery level (here, p = 95%) (i.e., pH*). In this example, R = 100% of the H* = 10 estimated haplotypes have been recovered for this species based on a sample size of only N = 100 specimens.

Unequal Haplotype Frequencies

Figures 6 and 7 show sample graphical output of the proposed haplotype accumulation curve simulation algorithm for a hypothetical species with N = 100 and H* = 10. All haplotypes occur with unequal frequency. Haplotypes 1–3 each have a frequency of 30%, while the remaining seven haplotypes each occur with a frequency of c. 1.4%.

## Set parameters for hypothetical species ##	
> N<-100	
> Hstar<-10	
> probs<-c(rep(0.30, 3), rep(0.10/7, 7)) # three dominant haplotypes each with 30% frequency	
### Run simulations ###	
> HACSObj<-HACHypothetical(N = N, Hstar = Hstar, probs = probs)	
> HAC.simrep(HACSObj)	
Simulating haplotype accumulation...	
—=========================================================— 100%	
—Measures of Sampling Closeness —	
Mean number of haplotypes sampled: 8.3291	
Mean number of haplotypes not sampled: 1.6709	
Proportion of haplotypes sampled: 0.83291	
Proportion of haplotypes not sampled: 0.16709	
Mean value of N*: 120.061	
Mean number of specimens not sampled: 20.06099	
Desired level of haplotype recovery has not yet been reached	
—====================================================— 100%	
—Measures of Sampling Closeness —	
Mean number of haplotypes sampled: 9.2999	
Mean number of haplotypes not sampled: 0.7001	
Proportion of haplotypes sampled: 0.92999	
Proportion of haplotypes not sampled: 0.07001	
Mean value of N*: 179.5718	
Mean number of specimens not sampled: 12.57182	
Desired level of haplotype recovery has not yet been reached	
—======================================================— 100%	
—Measures of Sampling Closeness —	
Mean number of haplotypes sampled: 9.5358	
Mean number of haplotypes not sampled: 0.4642	
Proportion of haplotypes sampled: 0.95358	
Proportion of haplotypes not sampled: 0.04642	
Mean value of N*: 188.7623	
Mean number of specimens not sampled: 8.762348	
Desired level of haplotype recovery has been reached	
———- Finished. ———-	
The initial guess for sampling sufficiency was N = 100 individuals	
The algorithm converged after 6 iterations and took 33.215 s	
The estimate of sampling sufficiency for p = 95% haplotype recovery is N∗ = 180 individuals	
( 95% CI [178.278–181.722])	
The number of additional specimens required to be sampled for p = 95% haplotype recovery is	
N* - N = 80 individuals	

Note that not all iterations are displayed above for the sake of brevity; only the first and last two iterations are given. With an initial guess of N = 100, only R = 83.3% of all H* = 10 observed haplotypes are recovered. The value of N* = 121 in the first iteration above serves as an improved initial guess of the true sampling sufficiency, which is an unknown quantity that is being estimated. This value is then fed back into the algorithm and the process is repeated until convergence is reached.

Figure 6 Initial graphical output of HAC.sim() for a hypothetical species having three dominant haplotypes.

(A) Specimens sampled; (B) Unique haplotypes. In this example, initially, only R = 83.3% of the H* = 10 estimated haplotypes have been recovered for this species based on a sample size of N = 100 specimens.

Figure 7 Final graphical output of HAC.sim() for a hypothetical species having three dominant haplotypes.

(A) Specimens sampled; (B) Unique haplotypes. In this example, upon convergence, R = 95.4% of the H* = 10 estimated haplotypes have been recovered for this species based on a sample size of N = 180 specimens.

Using Eq. (1), the improved sample size was calculated as N∗=100+1008.329110−8.3291=120.061. After one iteration, the curve has been extrapolated by an additional N∗ − Ni = 20.06099 individuals. Upon convergence, R = 95.4% of all observed haplotypes are captured with a sample size of N* = 180 specimens, with a 95% CI of 178.278–181.722. Given that N = 100 individuals have already been sampled, the number of additional specimens required is N∗ − N = 80 individuals. The user can verify that sample sizes close to that found by HACSim are needed to capture 95% of existing haplotype variation. Simply set N = N* = 180 and rerun the algorithm. The last iteration serves as a check to verify that the desired level of haplotype recovery has been achieved. The value of N* = 188.7623 that is outputted at this step can be used as a good starting guess to extrapolate the curve to higher levels of haplotype recovery to save on the number of iterations required to reach convergence. To do this, one simply runs HACHypothetical() with N = 189.

Application of HACSim to real species

Because the proposed iterative haplotype accumulation curve simulation algorithm simply treats haplotypes as numeric labels, it is easily generalized to any biological taxa and genetic loci for which a large number of high-quality DNA sequence data records is available in public databases such as BOLD. In the following examples, HACSim is employed to examine levels of standing genetic variation within animal species using 5′-COI.

Lake Whitefish (Coregonus clupeaformis)

An interesting case study on which to focus is that of Lake whitefish (Coregonus clupeaformis). Lake whitefish are a commercially, culturally, ecologically and economically important group of salmonid fishes found throughout the Laurentian Great Lakes in Canada and the United States, particularly to the Saugeen Ojibway First Nation (SON) of Bruce Peninsula in Ontario, Canada, as well as non-indigenous fisheries (Ryan & Crawford, 2014).

The colonization of refugia during Pleistocene glaciation is thought to have resulted in high levels of cryptic species diversity in North American freshwater fishes (Hubert et al., 2008; April et al., 2011; April et al., 2013a; April et al., 2013b). Overdyk et al. (2015) wished to investigate this hypothesis in larval Lake Huron lake whitefish. Despite limited levels of gene flow and likely formation of novel divergent haplotypes in this species, surprisingly, no evidence of deep evolutionary lineages was observed across the three major basins of Lake Huron despite marked differences in larval phenotype and adult fish spawning behaviour (Overdyk et al., 2015). This may be the result of limited sampling of intraspecific genetic variation, in addition to presumed panmixia (Overdyk et al., 2015). While lake whitefish represent one of the most well-studied fishes within BOLD, sampling effort for this species has nevertheless remained relatively static over the past few years. Thus, lake whitefish represent an ideal species for further exploration using HACSim.

In applying the developed algorithm to real species, sequence data preparation methodology followed that which is outlined in Phillips et al. (2015). Curation included the exclusion of specimens linked to GenBank entries, since those records without the BARCODE keyword designation lack appropriate metadata central to reference sequence library construction and management (Hanner, 2009). Our approach here was solely to assess comprehensiveness of single genomic sequence databases rather than incorporating sequence data from multiple repositories; thus, all DNA barcode sequences either originating from, or submitted to GenBank were not considered further. As well, the presence of base ambiguities and gaps/indels within sequence alignments can lead to bias in estimate haplotype diversity for a given species.

Currently (as of November 28, 2018), BOLD contains public (both barcode and non-barcode) records for 262 C. clupeaformis specimens collected from Lake Huron in northern parts of Ontario, Canada and Michigan, USA. Of the barcode sequences, N = 235 are of high quality (full-length (652 bp) and comprise no missing and/or ambiguous nucleotide bases). Haplotype analysis reveals that this species currently comprises H* = 15 unique COI haplotypes. Further, this species shows a highly-skewed haplotype frequency distribution, with a single dominant haplotype accounting for c. 91.5% (215/235) of all individuals (Fig. 8). The output of HACSim is displayed below.

### Run simulations ###	
> HACSObj<-HACReal()	
> HAC.simrep(HACSObj)	
Simulating haplotype accumulation...	
— =====================================================— 100%	
— Measures of Sampling Closeness —	
Mean number of haplotypes sampled: 11.0705	
Mean number of haplotypes not sampled: 3.9295	
Proportion of haplotypes sampled: 0.7380333	
Proportion of haplotypes not sampled: 0.2619667	
Mean value of N*: 318.4138	
Mean number of specimens not sampled: 83.4138	
Desired level of haplotype recovery has not yet been reached	
— =======================================================— 100%	
—Measures of Sampling Closeness —	
Mean number of haplotypes sampled: 13.8705	
Mean number of haplotypes not sampled: 1.1295	
Proportion of haplotypes sampled: 0.9247	
Proportion of haplotypes not sampled: 0.0753 Mean value of N*: 603.439	
Mean number of specimens not sampled: 45.43895	
Desired level of haplotype recovery has not yet been reached	
— ==========================================================— 100%	
—Measures of Sampling Closeness —	
Mean number of haplotypes sampled: 14.3708	
Mean number of haplotypes not sampled: 0.6292	
Proportion of haplotypes sampled: 0.9580533	
Proportion of haplotypes not sampled: 0.04194667	
Mean value of N*: 630.4451	
Mean number of specimens not sampled: 26.44507	
Desired level of haplotype recovery has been reached	
———- Finished. ———-	
The initial guess for sampling sufficiency was N = 235 individuals	
The algorithm converged after 8 iterations and took 241.235 s	
The estimate of sampling sufficiency for p = 95% haplotype recovery is N∗ = 604 individuals ( 95% CI: 601.504-606.496 )	
The number of additional specimens required to be sampled for p = 95% haplotype recovery is N* - N = 369 individuals	

From the above output, it is clear that current specimen sample sizes found within BOLD for C. clupeaformis are probably not sufficient to capture the majority of within-species COI haplotype variation. An initial sample size of N = 235 specimens corresponds to recovering only 73.8% of all H* = 15 unique haplotypes for this species (Fig. 9).

Figure 8 Initial haplotype frequency distribution for N= 235 high-quality lake whitefish (Coregonus clupeaformis) COI barcode sequences obtained from BOLD.

This species displays a highly-skewed pattern of observed haplotype variation, with Haplotype 1 accounting for c. 91.5% (215/235) of all sampled records.

Figure 9 Initial graphical output of HAC.sim() for a real species (Lake whitefish, C. clupeaformis) having a single dominant haplotype.

(A) Specimens sampled; (B) Unique haplotypes. In this example, initially, only R = 73.8% of the H* = 15 estimated haplotypes for this species have been recovered based on a sample size of N = 235 specimens. The haplotype frequency barplot is identical to that of Fig. 8.

A sample size of N* = 604 individuals (95% CI [601.504–606.496]) would likely be needed to observe 95% of all existing genetic diversity for lake whitefish (Fig. 10).

Figure 10 Final graphical output of HAC.sim() for Lake whitefish (C. clupeaformis) having a single dominant haplotype.

(A) Specimens sampled; (B) Unique haplotypes. Upon convergence, R = 95.8% of the H* = 15 estimated haplotypes for this species have been uncovered with a sample size of N = 604 specimens.

Since N = 235 individuals have been sampled previously, only N∗ − N = 369 specimens remain to be collected.

Deer tick (Ixodes scapularis)

Ticks, particularly the hard-bodied ticks (Arachnida: Acari: Ixodida: Ixodidae), are well-known as vectors of various zoonotic diseases including Lyme disease (Ondrejicka et al., 2014). Apart from this defining characteristic, the morphological identification of ticks at any lifestage, by even expert taxonomists, is notoriously difficult or sometimes even impossible (Ondrejicka, Morey & Hanner, 2017). Further, the presence of likely high cryptic species diversity in this group means that turning to molecular techniques such as DNA barcoding is often the only feasible option for reliable species diagnosis. Lyme-competent specimens can be accurately detected through employing a sensitive quantitative PCR (qPCR) procedure (Ondrejicka, Morey & Hanner, 2017). However, for such a workflow to be successful, wide coverage of within-species haplotype variation from across broad geographic ranges is paramount to better aid design of primer and probe sets for rapid species discrimination. Furthermore, the availability of large specimen sample sizes for tick species of medical and epidemiological relevance is necessary for accurately assessing the presence of the barcode gap.

Notably, the deer tick (Ixodes scapularis), native to Canada and the United States, is the primary carrier of Borrelia burgdorferi, the bacterium responsible for causing Lyme disease in humans in these regions. Because of this, I. scapularis has been the subject of intensive taxonomic study in recent years. For instance, in a recent DNA barcoding study of medically-relevant Canadian ticks, Ondrejicka, Morey & Hanner (2017) found that out of eight specimens assessed for the presence of B. burgdorferi, 50% tested positive. However, as only exoskeletons and a single leg were examined for systemic infection, the reported infection rate may be a lower bound due to the fact that examined specimens may still harbour B. burgdorferi in their gut. As such, this species is well-represented within BOLD and thus warrants further examination within the present study.

As of August 27, 2019, 531 5′-COI DNA barcode sequences are accessble from BOLD’s Public Data Portal for this species. Of these, N = 349 met criteria for high quality outlined in Phillips et al. (2015). A 658 bp MUSCLE alignment comprised H* = 83 unique haplotypes. Haplotype analysis revealed that Haplotypes 1–8 were represented by more than 10 specimens (range: 12–46; Fig. 11).

Simulation output of HACSim is depicted below.

### Run simulations ###	
> HACSObj<-HACReal()	
> HAC.simrep(HACSObj) Simulating haplotype accumulation...	
— ======================================================— 100%	
— Measures of Sampling Closeness —	
Mean number of haplotypes sampled: 65.3514	
Mean number of haplotypes not sampled: 17.6486	
Proportion of haplotypes sampled: 0.7873663	
Proportion of haplotypes not sampled: 0.2126337	
Mean value of N*: 443.2499	
Mean number of specimens not sampled: 94.24988	
Desired level of haplotype recovery has not yet been reached	
— ============================================— 100%	
—Measures of Sampling Closeness —	
Mean number of haplotypes sampled: 78.3713	
Mean number of haplotypes not sampled: 4.6287	
Proportion of haplotypes sampled: 0.9442325	
Proportion of haplotypes not sampled: 0.05576747	
Mean value of N*: 802.7684	
Mean number of specimens not sampled: 44.76836	
Desired level of haplotype recovery has not yet been reached	
— =========================================================— 100%	
—Measures of Sampling Closeness —	
Mean number of haplotypes sampled: 79.2147	
Mean number of haplotypes not sampled: 3.7853	
Proportion of haplotypes sampled: 0.954394	
Proportion of haplotypes not sampled: 0.04560602	
Mean value of N*: 841.3716	
Mean number of specimens not sampled: 38.37161	
Desired level of haplotype recovery has been reached	
———- Finished. ———-	
The initial guess for sampling sufficiency was N = 349 individuals	
The algorithm converged after 8 iterations and took 1116.468 s	
The estimate of sampling sufficiency for p = 95% haplotype recovery is N∗ = 803 individuals ( 95% CI: 801.551-804.449 )	
The number of additional specimens required to be sampled for p = 95% haplotype recovery is N* - N = 454 individuals	

The above results clearly demonstrate the need for increased specimen sample sizes in deer ticks. With an initial sample size of N = 349 individuals, only 78.7% of all observed haplotypes are recovered for this species (Fig. 12). N* = 803 specimens (95% CI: 801.551-804.449) is necessary to capture at least 95% of standing haplotype variation for I. scapularis (Fig. 13) . Thus, a further N* − N = 454 specimens are required to be collected.

Figure 11 Initial haplotype frequency distribution for N= 349 high-quality deer tick (Ixodes scapularis) COI barcode sequences obtained from BOLD.

In this species, Haplotypes 1-8 account for c. 51.3% (179/349) of all sampled records.

Figure 12 Initial graphical output of HAC.sim() for a real species (Deer tick, I. scapularis) having eight dominant haplotypes.

In this example, initially, only R = 78.7% of the H* = 83 estimated haplotypes for this species have been recovered based on a sample size of N = 349 specimens. The haplotype frequency barplot is identical to that of Fig. 11.

Figure 13 Final graphical output of HAC.sim() for deer tick (I scapularis) having eight dominant haplotypes.

Upon convergence, R = 95.4% of the H* = 83 estimated haplotypes for this species have been uncovered with a sample size of N = 803 specimens.

Scalloped hammerhead (Sphyrna lewini)

Sharks (Chondrichthyes: Elasmobranchii: Selachimorpha) represent one of the most ancient extant lineages of fishes. Despite this, many shark species face immediate extinction as a result of overexploitation, together with a unique life history (e.g., K-selected, predominant viviparity, long gestation period, lengthy time to maturation) and migration behaviour (Hanner, Naaum & Shivji, 2016). A large part of the problem stems from the increasing consumer demand for, and illegal trade of, shark fins, meat and bycatch on the Asian market. The widespread, albeit lucrative practice of “finning”, whereby live sharks are definned and immediately released, has led to the rapid decline of once stable populations (Steinke et al., 2017). As a result, numerous shark species are currently listed by the International Union for the Conservation of Nature (IUCN) and the Convention on International Trade in Endangered Species of Wild Fauna and Flora (CITES). Interest in the molecular identification of sharks through DNA barcoding is multifold. The COI reference sequence library for this group remains largely incomplete. Further, many shark species exhibit high intraspecific distances within their barcodes, suggesting the possibility of cryptic species diversity. Instances of hybridization between sympatric species has also been documented. As establishing species-level matches to partial specimens through morphology alone is difficult, and such a task becomes impossible once fins are processed and sold for consumption or use in traditional medicine, DNA barcoding has paved a clear path forward for unequivocal diagnosis in most cases.

The endangered hammerheads (Family: Sphyrnidae) represent one of the most well-sampled groups of sharks within BOLD to date. Fins of the scalloped hammerhead (Sphyrna lewini) are especially highly prized within IUU (Illegal, Unregulated, Unreported) fisheries due to their inclusion as the main ingredient in shark fin soup.

As of August 27, 2019, 327 S. lewini specimens (sequenced at both barcode and non-barcode markers), collected from several Food and Agriculture Organization (FAO) regions, including the United States, are available through BOLD’s Public Data Portal. Of these, all high-quality records (N = 171) were selected for alignment in MEGA7 and assessment via HACSim. The final alignment was found to comprise H* = 12 unique haplotypes, of which three were represented by 20 or more specimens (range: 28–70; Fig. 14).

HACSim results are displayed below.

### Run simulations ###	
> HACSObj<-HACReal()	
> HAC.simrep(HACSObj)	
Simulating haplotype accumulation...	
— ======================================================— 100%	
—Measures of Sampling Closeness —	
Mean number of haplotypes sampled: 9.9099	
Mean number of haplotypes not sampled: 2.0901	
Proportion of haplotypes sampled: 0.825825	
Proportion of haplotypes not sampled: 0.174175	
Mean value of N*: 207.0657	
Mean number of specimens not sampled: 36.06566	
Desired level of haplotype recovery has not yet been reached	
===================================================== 100%	
—Measures of Sampling Closeness —	
Mean number of haplotypes sampled: 11.3231	
Mean number of haplotypes not sampled: 0.6769	
Proportion of haplotypes sampled: 0.9435917	
Proportion of haplotypes not sampled: 0.05640833	
Mean value of N*: 413.3144	
Mean number of specimens not sampled: 23.31438	
Desired level of haplotype recovery has not yet been reached	
===================================================== 100%	
—Measures of Sampling Closeness —	
Mean number of haplotypes sampled: 11.4769	
Mean number of haplotypes not sampled: 0.5231	
Proportion of haplotypes sampled: 0.9564083	
Proportion of haplotypes not sampled: 0.04359167	
Mean value of N*: 432.8695	
Mean number of specimens not sampled: 18.8695	
Desired level of haplotype recovery has been reached	
———- Finished. ———-	
The initial guess for sampling sufficiency was N = 171 individuals	
The algorithm converged after 9 iterations and took 174.215 s	
The estimate of sampling sufficiency for p = 95% haplotype recovery is N* = 414 individuals ( 95% CI: 411.937-416.063 )	
The number of additional specimens required to be sampled for p = 95% haplotype recovery is N* - N = 243 individuals	

Simulation output suggests that only 82.6% of all unique haplotypes for the scalloped hammerhead have likely been recovered (Fig. 15) with a sample size of N = 171.

Figure 14 Initial haplotype frequency distribution for N= 171 high-quality scalloped hammerhead (Sphyrna lewini) COI barcode sequences obtained from BOLD.

In this species, Haplotypes 1–3 account for c. 87.7% (150/171) of all sampled records.

Figure 15 Initial graphical output of HAC.sim() for a real species (Scalloped hammerhead, S. lewini) having three dominant haplotypes.

In this example, initially, only R = 82.6% of the H* = 12 estimated haplotypes for this species have been recovered based on a sample size of N = 171 specimens. The haplotype frequency barplot is identical to that of Fig. 14.

Figure 16 Final graphical output of HAC.sim() for scalloped hammerhead (S. lewini) having three dominant haplotypes.

Upon convergence, R = 95.6% of the H* = 12 estimated haplotypes for this species have been uncovered with a sample size of N = 414 specimens.

Further, HACSim predicts that N* = 414 individuals (95% CI [411.937–416.063]) probably need to be randomly sampled to capture the majority of intraspecific genetic diversity within 5′-COI (Fig. 16). Since 171 specimens have already been collected, this leaves an additional N* − N = 243 individuals which await sampling.

Discussion

Initializing HACSim and overall algorithm behaviour

The overall stochastic behaviour of HACSim is highly dependent on the number of permutations used upon algorithm initialization. Provided that the value assigned to the perms argument is set high enough, numerical results ouputted by HACSim will be found to be quite consistent between consecutive runs whenever all remaining parameter values remain unchanged. It is crucial that perms not be set to too low a value to prevent the algorithm from getting stuck at local maxima and returning suboptimal solutions. This is a common situation with popular optimization algorithms such as hill-climbing. Attention therfore must be paid to avoid making generalizations based on algorithm performance and obtained simulation results (Spall, 2012).

In applying the present method to simulated species data, it is important that selected simulation parameters are adequately reflective of those observed for real species. Thus, initial sample sizes should be chosen to cover a wide range of values based on those currently observed within BOLD. Such information can be gauged through examining species lists associated with BOLD records, which are readily accessible through Linnean search queries within the Taxonomy browser.

As with any iterative numerical algorithm, selecting good starting guesses for initialization is key. While HACSim is globally convergent (i.e., convergence is guaranteed for any value of N ≥ H*), a good strategy when simulating hypothetical species is to start the algorithm by setting N = H*. In this way, the observed fraction of haplotypes found, R, will not exceed the desired level of haplotype recovery p, and therefore lead to overestimation of likely required specimen sample sizes. Setting N high enough will almost surely result in R exceeding p. Thus, arbitrarily large values of N may not be biologically meaningful or practical. However, in the case of hypothetical species simulation, should initial sample sizes be set too high, such that R > p, a straightforward workaround is to observe where the dashed horizontal line intersects the final haplotype accumulation curve (i.e., not the line the touches the curve endpoint). The resulting value of N at this point will correspond with p quite closely. This can be seen in Fig. 5, where an eyeball guess just over N* = 20 individuals is necessary to recover p = 95% haplotype variation. A more reliable estimate can be obtained through examining the dataframe “d” outputted once the algorithm has halted (via envr$d). In this situation, simply look in the row corresponding to pH* ≥ 0.95(10) ≥ 9.5. The required sample size is the value given in the first column (specs). This is accomplished via the R code envr$d[which(envr$d$means > = envr$p * envr$Hstar), ][1, 1].

The novelty of HACSim is that it offers a systematic means of estimating likely specimen sample sizes required to assess intraspecific haplotype diversity for taxa within large-scale genomic databases like GenBank and BOLD. Estimates of sufficient sampling suggested by our algorithm can be employed to assess barcode coverage within existing reference sequence libraries and campaign projects found in BOLD. While comparison of our method to already-established ones is not yet possible, we anticipate that HACSim will nevertheless provide regulatory applications with an unprecedented view and greater understanding of the state of standing genetic diversity (or lack thereof) within species.

Additional capabilities and extending functionality of HACSim

In this paper, we illustrate the application of haplotype accumulation curves to the broad assessment of species-level genetic variation. However, HACSim is quite flexible in that one can easily explore likely required sample sizes at higher taxonomic levels (e.g., order, family, genus) or specific geographic regions (e.g., salmonids of the Great Lakes) with ease. Such applicability will undoubtedly be of interest at larger scales (i.e.. entire genomic sequence libraries). For instance, due to evidence of sampling bias in otherwise densely-sampled taxa housed in BOLD (e.g., Lepidoptera), D’Ercole et al. (J. D’Ercole, 2019, unpublished data) wished to assess whether or not intraspecific haplotype variation within butterfly species remains unsampled. To test this, the authors employed HACSim to examine sampling comprehensiveness for species comprising a large barcode reference library for North American butterflies spanning 814 species and 14,623 specimens.

We foresee use of HACSim being widespread within the DNA barcoding community. As such, improvements to existing code in terms of further optimization and algorithm runtime, as well as implementation of new methods by experienced R programmers in the space of DNA-based taxonomic identification, seems bright.

Potential extensions of our algorithm include support for the exploration of genetic variation at the Barcode Index Number (BIN) level (Ratnasingham & Hebert, 2013), as well as high-throughput sequencing (HTS) data for metabarcoding and environmental DNA (eDNA) applications. Such capabilities are likely to be challenging to implement at this stage until robust operational taxonomic unit (OTU) clustering algorithms are developed (preferably in R). One promising tool in this regard for barcoding of bulk samples of real species and mock communities of known species composition is JAMP (Just Another Metabarcoding Pipeline) devised for use in R by Elbrecht and colleagues (Elbrecht et al., 2018). JAMP includes a sequence read denoising tool that can be used to obtain haplotype numbers and frequency information (H* and probs). However, because JAMP relies on third-party software (particularly USEARCH (Edgar, 2010) and VSEARCH (Rognes et al., 2016)), it cannot be integrated within HACSim itself and will thus have to be used externally. In extending HACSim to next-generation space, two issues arise. First, it is not immediately clear how the argument N, is to be handled since multiple reads could be associated with single individuals. That is, unlike in traditional Sanger-based sequencing, there is not a one-to-one correspondence between specimen and sequence (Wares & Pappalardo, 2015; Adams et al., 2019). Second, obtaining reliable haplotype information from noisy HTS datasets is challenging without first having strict quality filtering criteria in place to minimize the occurrence of rare, low-copy sequence variants which may reflect artifacts stemming from the Polymerase Chain Reaction (PCR) amplification step or sequencing process (Elbrecht et al., 2018; Braukmann et al., 2019; Turon et al., 2019). Turning to molecular population genetics theory might be the answer (Adams et al., 2019). Wares & Pappalardo (2015) suggest three different approaches to estimating the number of specimens of a species that may have contributed to a metabarcoding sample: (1) use of prior estimates of haplotype diversity, together with observed number of haplotypes; (2) usage of Ewens’ sampling formula (Ewens, 1972) along with estimates of Watterson’s θ (not to be confused with the θ denoting true sampling sufficency herein) (Watterson, 1975), as well as total number of sampled haplotypes; and (3) employment of an estimate of θ and the number of observed variable sites (S) within a multiple sequence alignment. A direct solution we propose might be to use sequencing coverage/depth (i.e., the number of sequence reads) as a proxy for number of individuals. The outcome of this would be an estimate of the mean/total number of sequece reads required for maximal haplotype recovery. However, the use of read count as a stand-in for number of specimens sampled would require the unrealistic assumption that all individuals (i.e., both alive and dead) shed DNA into their environment at equal rates. The obvious issue with extending HACSim to handle HTS data is computing power, as such data typically consists of millions of reads spanning multiple gigabytes of computer memory.

Summary

Here, we introduced a new statistical approach to assess specimen sampling depth within species based on existing gene marker variation found in public sequence databanks such as BOLD and GenBank. HACSim is both computationally efficient and easy to use. We show utility of our proposed algorithm through both hypothetical and real species genomic sequence data. For real species (here, lake whitefish, deer tick and scalloped hammerhead), results from HACSim suggest that comprehensive sampling for species comprising large barcode libraries within BOLD, such as Actinopterygii, Arachnida and Elasmobranchii is far from complete. With the availability of HACSim, appropriate sampling guidelines based on the amount of potential error one is willing to tolerate can now be established. For the purpose of addressing basic questions in biodiversity science, the employment of small taxon sample sizes may be adequate; however, this is not the case for regulatory applications, where greater than 95% coverage of intraspecific haplotype variation is needed to provide high confidence in sequence matches defensible in a court of law.

Of immediate interest is the application of our method to other ray-finned fishes, as well as other species from deeply inventoried taxonomic groups such as Elasmobranchii (e.g., sharks), Insecta (e.g., Lepidoptera, Culicidae (mosquitoes)), Arachnida (e.g., ticks) and Chiroptera (bats) that are of high conservation, medical and/or socioeconomic importance. Although we explicitly demonstrate the use of HACSim through employing COI, it would be interesting to extend usage to other barcode markers such as the ribulose-1,5-bisphosphate carboxylase/oxygenase large subunit (rbcL) and maturase K (matK) chloroplast genes for land plants, as well as the nuclear internal transcribed spacer (ITS) marker regions for fungi. The application of our method to non-barcode genes routinely employed in specimen identification like mitochondrial cytochrome b (cytb) in birds for instance (Baker, Sendra Tavares & Elbourne, 2009; Lavinia et al., 2016), nuclear rhodopsin (rho) for marine fishes (Hanner et al., 2011) or the phosphoenolpyruvate carboxykinase (PEPCK) nuclear gene for bumblebees (Williams et al., 2015) is also likely to yield interesting results since sequencing numerous individuals at several different genomic markers can often reveal evolutionary patterns not otherwise seen from employing a single-gene approach (e.g., resolution of cryptic species or confirmation/revision of established taxonomic placements) (Williams et al., 2015).

While it is reasonable that HACSim can be applied to genomic regions besides 5′-COI, careful consideration of varying rates of molecular evolution within rapidly-evolving gene markers and the effect on downsteam inferences is paramount, as is sequence quality. Previous work in plants (Genus: Taxus) by Liu et al. (2012) has found evidence of a correlation between mutation rate and required specimen sampling depth: genes evolving at faster rates will likely require larger sample sizes to estimate haplotype diversity compared to slowly-evolving genomic loci. We simply focused on 5′-COI because it is by far the most widely sequenced mitochondrial locus for specimen identification, owing to its desirable biological properties as a DNA barcode for animal taxa and because it has an associated data standard to help filter out poor-quality data. (Phillips, Gillis & Hanner, 2019). However, it should be noted that species diagnosis using COI and other barcode markers is not without its challenges. While COI accumulates variation at an appreciable rate, certain taxonomic groups are not readily distinguished on the basis of their DNA barcodes (e.g., the so-called “problem children”, such as Cnidaria, which tend to lack adequate sequence divergence (Bucklin, Steinke & Blanco-Bercial, 2011)). Other taxa, like Mollusca, are known to harbour indel mutations (Layton, Martel & Hebert, 2014). Introns within Fungi greatly complicate sequence alignment (Min & Hickey, 2007). Thus, users of HACSim must exercise caution in interpreting end results with other markers, particularly those which are not protein-coding.

It is necessary to consider the importance of sampling sufficiency as it pertains to the myriad regulatory applications of specimen identification established using DNA barcoding (e.g., combatting food fraud) in recent years. It since has become apparent that the success of such endeavours is complicated by the ever-evolving state of public reference sequence libraries such as those found within BOLD, in addition to the the inclusion of questionable sequences and lack of sufficent metadata for validation purposes in other genomic databases like GenBank (e.g., Harris (2003)). Dynamic DNA-based identification systems may produce multiple conflicting hits to otherwise corresponding submissions over time. This unwanted behaviour has led to a number of regulatory agencies creating their own static repositories populated with expertly-identified sequence records tied to known voucher specimens deemed fit-for-purpose for molecular species diagnosis and forensic compliance (e.g., the United States Food and Drug Administration (USFDA)’s Reference Standard Sequence Library (RSSL) employed to identify unknown seafood samples from species of high socioeconomic value). While such a move has partially solved the problem of dynamism inherent in global sequence databases, there still remains the issue of low sample sizes that can greatly inflate the perception of barcode gaps between species. Obtaining adequate representation of standing genetic variation, both within and between species, is therefore essential to mitigating false assignments using DNA barcodes. To this end, we propose the use of HACSim to assess the degree of saturation of haplotype accumulation curves to aid regulatory scientists in rapidly and reliably projecting likely sufficient specimen sample sizes required for accurate matching of unknown queries to known Linnean names.

A defining characteristic of HACSim is its convergence behaviour: the method converges to the desired level of haplotype recovery p for any initial guess N specified by the user. Based on examples explored herein, it appears likely that already-sampled species within repositories like BOLD are far from being fully characterized on the basis of existing haplotype variation. In addition to this, it is important to consider the current limitations of our algorithm. We can think of only one: it must be stressed that appropriate sample size trajectories are not possible for species with only single representatives within public DNA sequence databases because haplotype accumulation is unachievable with only one DNA sequence and/or a single sampled haplotype. Hence, HACSim can only be applied to species with at least two sampled specimens. Thus, application of our method to assess necessary sample sizes for full capture of extant haplotype variation in exceedingly rare or highly elusive taxa is not feasible. Despite this, we feel that HACSim can greatly aid in accurate and rapid barcode library construction necessary to thoroughly appreciate the diversity of life on Earth.

Conclusions

Herein, a new, easy-to-use R package was presented that can be employed to estimate intraspecific sample sizes for studies of genetic diversity assessment, with a particular focus on animal DNA barcoding using the COI gene. HACSim employs a novel nonparametric stochastic iterative extrapolation algorithm with good convergence properties to generate haplotype accumulation curves. Because our approach treats species’ haplotypes as numeric labels, any genomic locus can be targeted to probe levels of standing genetic variation within multicellular taxa. However, we stress that users must exercise care when dealing with sequence data from non-coding regions of the genome, since these are likely to comprise sequence artifacts such as indels and introns, which can both hinder successful sequence alignment and lead to overestimation of existing haplotype variation within species. The application of our method to assess likely required sample sizes for both hypothetical and real species produced promising results. We argue the use of HACSim will be of broad interest in both academic and industry settings, most notably, regulatory agencies such as the Canadian Food Inspection Agency (CFIA), Agriculture and Agri-Food Canada (AAFC), United States Department of Agriculture (USDA), Public Health Agency of Canada (PHAC) and the USFDA. While HACSim is an ideal tool for the analysis of Sanger sequencing reads, an obvious next step is to extend usability to Next-Generation Sequencing (NGS), especially HTS applications. With these elements in place, even the full integration of HACSim to assess comprehensiveness of taxon sampling within large sequence databases such as BOLD seems like a reality in the near future.

Supplemental Information

Supplemental Information 1 5′-COI DNA barcode sequence alignment for lake whitefish (Coregonus clupeaformis)

Click here for additional data file.

Supplemental Information 2 5′-COI DNA barcode sequence alignment for deer tick (Ixodes scapularis)

Click here for additional data file.

Supplemental Information 3 5′-COI DNA barcode sequence alignment for scalloped hammerhead shark (Sphyrna lewini)

Click here for additional data file.

Supplemental Information 4 Additional capabilities of HACSim

Click here for additional data file.

The authors wish to extend thanks to Robert (Rob) Young for helpful discussions and providing critical feedback on an earlier draft of this manuscript that greatly improved its flow and overall readability. We acknowledge that the University of Guelph resides on the ancestral lands of the Attawandaron people and the treaty lands and territory of the Mississaugas of the Credit. We recognize the significance of the Dish with One Spoon Covenant to this land and offer our respect to our Anishinaabe, Haudenosaunee and Métis neighbours as we strive to strengthen our relationships with them.

Additional Information and Declarations

Competing Interests

Author Contributions

Data Availability

The authors declare there are no competing interests.

Jarrett D. Phillips conceived and designed the experiments, performed the experiments, analyzed the data, contributed reagents/materials/analysis tools, prepared figures and/or tables, performed the computation work, authored or reviewed drafts of the paper, approved the final draft.

Steven H. French conceived and designed the experiments, performed the computation work, authored or reviewed drafts of the paper, approved the final draft.

Robert H. Hanner and Daniel J. Gillis conceived and designed the experiments, authored or reviewed drafts of the paper, approved the final draft.

The following information was supplied regarding data availability:

The aligned and trimmed DNA barcodes are available at FigShare: Phillips, Jarrett (2019): Coregonus clupeaformis 5′-COI DNA barcode sequences. figshare. Dataset. 10.6084/m9.figshare.8870804.v1.

Phillips, Jarrett (2019): Ixodes scapularis 5′-COI DNA barcode sequences. figshare. Dataset. 10.6084/m9.figshare.10006595.v1.

Phillips, Jarrett (2019): Sphyrna lewini 5′-COI DNA barcode sequences. figshare. Dataset. 10.6084/m9.figshare.10006598.v1.

A stable version of the algorithm is available on GitHub: https://github.com/jphill01/HACSim.R, along with a detailed README on how to run code.

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
