# Peer review of "HACSim: an R package to estimate intraspecific sample sizes for genetic diversity assessment using haplotype accumulation curves"

_PeerJ Computer Science, doi:10.7717/peerj-cs.243_

## Round 0.1 · original submission · Major Revisions

· Academic Editor

Major Revisions

Based on the reviewers' comments and my own evaluation, I think this work is nearly ready for publication after the following revisions.

(1) Please present a more accurate title of the manuscript. For example, in the title, the authors should tell the readers “the estimate of sample size”. In the current title, it is complex and long. For those readers with the background of computer science, the title may not clear enough. Therefore, I suggest the author to modify the title to be more accurate.

(2) In the section of Introduction, please clearly indicate the main contributions made in this paper. For example, use the following sentences: “The main contributions made in the paper can be summarized as follows”. The work is nice, but please clearly tell the readers your main contributions directly.

(3) In the section of Method, please first present a figure of the flowchart to describe the process of the proposed algorithm HACSim. This may help readers be easier to understand the algorithm HACSim.

(4) In subsection 2.1.1 Motivation, I suggest authors move the motivation to the proper place in the section of Introduction. In the Method section, the authors only need to tell the readers all the details of the proposed algorithm. But, the reason why proposing the algorithm should be placed in the section of Introduction.

(5) The results are sufficient. But I suggest the authors present the code using the standard format, e.g., the listing in Latex. Currently, the code is not format or standard.

(6) Please additionally present a clear and short section of Conclusions.

(7) Please improve the quality of all your figures using the software Microsoft Vision, or Adobe Illustrator, or Origin.

Reviewer 1 ·

Basic reporting

no comment

Experimental design

no comment

Validity of the findings

no comment

Additional comments

Based on previous series of work in related field(s), the authors have introduced a new simulation algorithm for calculating sample sized needed in DNA barcoding studies. The manuscript is well written with clear logic and enough cited literatures.

Considering the progresses in next-generation sequencing and species delimitation, I suggest that the authors consider more complex biological targets such as multiple loci or genomic data (e.g., SNP) rather than the DNA barcode region (i.e., 648 bp of COI gene) in future work.

·

Basic reporting

Summary/gist: The authors' hypothesis stems from assessing the genetic variation using the molecular techniques such as DNA barcoding which is widely employed in the 'Now Generation Sequencing' era. The authors give a strong emphasis on existing approaches and present a stochastic model using Monte Carlo simulations by employing an iterative algorithm. The algorithm, as the authors state, appears to be a fine-tuned approach based on Philips et al., 2015. Furthermore, they study the effect of haplotype frequences in cytochrome c oxidase subunit 1, a mitochondrial 5'UTR sparsed region. Finally they present the benchmark/proof-of-concept in fishes besides taking certain hypothetical examples.

Strengths: The work is absolutely novel, the need of the hour, and a subject of interest coalescing statistical geneticists and biologists towards development of barcoding markers. The manuscript is written well with good review and statistical interpretation.

Weaknesses: Sometimes, the flow is too much carried away and the flow could have been better with a methodological flowchart.

The R algorithm could be linked with github/bitbucket repositories.

Reinstantiation of codes, if any must be mentioned as per the Minimum Information About Bioinformatics Investigation ( MIABI) guidelines.

Experimental design

The methodology of Philips et al., brings an overestimates/overfitting of the samples and this brings up the rationale where the authors develop a more iterative/stochastic model in the form of HACSim. The figure 1 represents the schematic representation of this.

The haplotype variation could be best set for Metagenomic samples. For examples, binning the reads into haplotypes, and further into strains would allow us to define the specific barcodes at species level. Although the authors have subtly mentioned the role of COI and employed certain frequencies, they could have considered 'metagenomes' for that barcoding is a big problem for assessing the metagenomic samples a sthey are in turn bound by tags and primer sequences for identifying the accurate sequences.

Could this algorithm identify and recover such haplotypes?

Could the authors use a non-stochastic model to run their code and see how it behaves? Although it sounds unorthodoxical, this would set an assumption to depict inherently stochastic model.

Assuming that the haplotype frequencies are not equal, how similar would be the haplotype frequencies at a certainty/random?

Validity of the findings

I am bit intrigued by the reason why the authors chose Lake whitefish excepting the reason de tre and the association with "species diversity." Aren't there any other datasets to measure species diversity? The gene flow hypothesis could be measured at microbial level as well.

How would this HACSim be associated, for example on rRNA reads which has distinct biological properties as well.

---

## Round 0.2 · accepted · Accept

· Academic Editor

Accept

I am also happy that the authors have significantly improved the quality of their manuscript according to the reviewers’ and my comments. Here, I still suggest the authors use the revised high-quality figures when preparing the final files for publication.

·

Basic reporting

I am happy and convinced with teh changes rendered by the authors. Thank you

Experimental design

None

Validity of the findings

None

Additional comments

I am happy and convinced with teh changes rendered by the authors. Thank you